# Time-Efficient SNR Optimization of WMS-Based Gas Sensor Using a Genetic Algorithm

**DOI:** 10.3390/s24061842

**Published:** 2024-03-13

**Authors:** Filip Musiałek, Dariusz Szabra, Jacek Wojtas

**Affiliations:** Institute of Optoelectronics, Military University of Technology, 2 Kaliskiego Str., 00-908 Warsaw, Poland

**Keywords:** laser absorption spectroscopy, SNR optimization, WMS, methane sensor, artificial intelligence, genetic algorithm, LWIR

## Abstract

This paper presents the description of the wavelength modulation spectroscopy (WMS) experiment, the parameters of which were established by use of the Artificial Intelligence (AI) algorithm. As a result, a significant improvement in the signal power to noise power ratio (SNR) was achieved, ranging from 1.6 to 6.5 times, depending on the harmonic. Typically, optimizing the operation conditions of WMS-based gas sensors is based on long-term simulations, complex mathematical model analysis, and iterative experimental trials. An innovative approach based on a biological-inspired genetic algorithm (GA) and custom-made electronics for laser control is proposed. The experimental setup was equipped with a 31.23 m Heriott multipass cell, software lock-in, and algorithms to control the modulation process of the quantum cascade laser (QCL) operating in the long-wavelength-infrared (LWIR) spectral range. The research results show that the applied evolutionary approach can efficiently and precisely explore a wide range of WMS parameter combinations, enabling researchers to dramatically reduce the time needed to identify optimal settings. It took only 300 s to test approximately 1.39 × 10^32^ combinations of parameters for key system components. Moreover, because the system is able to check all possible component settings, it is possible to unquestionably determine the operating conditions of WMS-based gas sensors for which the limit of detection (LOD) is the most favorable.

## 1. Introduction

Nowadays, tunable laser absorption spectroscopy (TLAS) plays a significant role in the detection of miscellaneous gases as well as in the measurement of their concentration, temperature, or pressure [1,2,3]. This method is widely used for the detection of hazardous substances in industry, mining, aviation, border services, and military [4,5]. In addition, laser absorption spectroscopy contributes to the fight against global warming and terrorism and is also used to detect disease biomarkers [6,7,8].

Applying laser wavelength modulation can increase the signal-to-noise ratio (SNR) and, hence, the sensitivity of the TLAS systems. The theoretical basis, mathematical, and simulation models of wavelength modulation spectroscopy (WMS) have been very well described and practically tested in many papers by various authors [9,10,11]. SNR improvement in WMS is mainly achieved by narrowing the sensor’s bandwidth to the modulation frequency. This is possible by applying phase-sensitive detection using Lock-In amplifiers. Even-order harmonics magnitude depends on the shape of the absorption lines and is proportional to the concentration, temperature, and pressure of detected gases. Therefore, these demodulated signals are used to determine the properties of detected substances. On the other hand, odd-order harmonics are very often used for laser power fluctuation normalization [12]. According to the Lambert–Beer law, to increase the absorption of the laser radiation by gas molecules for a given wavelength, the optical pathlength may be increased. Therefore, multi-pass absorption cells are also used in TLAS. While this solution allows for an increase in the sensitivity of the system, it often introduces additional distortions and optical interferences. Moreover, the absorption signal can be enhanced by optimizing several laser control parameters [11].

In general, to achieve high sensitivity and selectivity of the TLAS sensor, it is necessary to carefully select both its spectral and modulating parameters. Firstly, it is required to accurately adjust the spectral characteristics of the laser, optical system, and photodetector to the selected gas absorption band. The next important step should be to select the laser modulation parameters. It was theoretically and experimentally proved that the modulation index affects the peak height of harmonics and, thus, the sensitivity of the system. For example, for second harmonics (2f), the optimal value of the modulation index is around value 2.2 [13]; however, this value depends on the temperature and pressure of the gas [14]. Theoretically, increasing the modulation frequency should cause a better SNR due to the minimization of the 1/f noise [15]. On the other hand, the choice of modulation frequency is limited by the current driver, laser, and photodetector frequency response, as well as the sampling frequency of the ADC. While the tuning frequency value is dictated by the required gas sensor response speed, it must be significantly lower than the modulation frequency. Therefore, in most WMS applications, the laser is modulated with kHz-order and tuned with Hz-order frequencies. Moreover, tuning signal amplitude (most cases ramp) determines the tuning range of the laser wavelength in relation to the absorption line. Most often, the tuning amplitude is several times wider than the FWHM of the absorption line. However, when selecting modulation values, other phenomena should also be considered, such as optical fringes [16], residual amplitude modulation (RAM), and other noise sources [10]. Due to the complicated relationships between the frequency and intensity parameters of the modulation and laser scanning, it poses a quite big challenge to find a global optimum for all modulation parameters. Moreover, there are generally no explicit mathematical expressions that describe all dependence of the WMS signal on the laser control parameters [17]. Therefore, modulation parameters are determined experimentally for the specific configuration of the sensor as well as absorption characteristics. This, however, may be very time-consuming, requiring many investigations of system components and gas parameter measurements. However, above all, expert knowledge is needed when developing and launching the experiment.

In this paper, a new method for optimizing the parameters of WMS gas analyzers using genetic algorithms is proposed. Previously published articles about optimizing the performance of WMS sensors have primarily focused on novel modulation methods [18], simulation-based exploration of the optimal modulation index [19], or theoretical examination of various noise reduction algorithms [20]. However, to the best of our knowledge, no one has yet undertaken the task of developing an automatic, time-efficient method for optimizing WMS sensor parameters. The advantage of the proposed method is its universality because it does not depend on the laser wavelength, type, and parameters of the detected gas. It ensures the optimal operating parameters within a defined range and their adaptive adjustment to changing measurement conditions, ensuring the highest sensor performance. The developed solution was demonstrated in the example of a methane sensor using an absorption line located in LWIR [21].

## 2. Materials and Methods

One of the most important parameters describing the performance of a gas sensor is LOD. Therefore, to ensure the universality of the proposed method, it was decided to measure the SNR, which allows for a comprehensive assessment of the effectiveness of the proposed algorithm, as it is determined by the parameters of the sensor system elements and has a direct impact on LOD. According to the International Union of Pure and Applied Chemistry (IUPAC), LOD is the smallest concentration *C_L_* of gas that has a signal statistically significantly larger than the signal arising from the repeated measurements of a blank test [22,23]. In the WMS technique, measurements of the gas concentration are determined by measuring the voltage *U* at the output of a phase-sensitive amplifier. The sensor must then be usually calibrated using the calibration curve, which is prepared using a series of signal measurements for different gas concentrations. Consequently, the LOD, expressed as the concentration, can be derived from the smallest measure *U_L_* that can be measured using the following equation:(1)$$U_{L}=U_{b,m}+3\sigma_{b},$$
where *U_b_*_,*m*_ is mean voltage of the blank measures, and *σ_b_* is the standard deviation of the blank measures. The minimum measured concentration *C_L_* can be calculated using sensitivity *S* defined as the slope of the sensor’s analytical calibration curve:(2)$$C_{L}=\frac{\left(U_{L}-U_{b,m}\right)}{S}=\frac{3\sigma_{b}}{S}.$$

The standard deviation of the blank measurements is registered at the output of the phase-sensitive amplifier as the accumulative sensor noise. Assuming a typical linear curve of calibration plot for low values of *C*, sensitivity can be estimated as a derivative of harmonic amplitude *A_nf_* and concentration *C_x_* of gas in the absorption cell. Therefore, with the premise of the calibration curve having a C-intercept equal to zero, we can write a simplified equation:(3)$$C_{L}\approx \frac{3\sigma_{b}}{\frac{A_{nf}}{C_{x}}}\approx \frac{3C_{x}}{SNR}$$

Upon analyzing the above equation, it is observed that when the SNR of the WMS sensor, detecting a stable gas sample, is improved from the initial value *SNR*_0_ to the optimized value *SNR*_1_, the LOD will be enabled to improve from *C_L_*_0_ to *C_L_*_1_ in the following manner:(4)$$C_{L1}=\frac{{SNR}_{0}}{{SNR}_{1}}C_{L0}$$

### 2.1. Experimental Setup

A demonstration of the optimization method was performed using the long-wave infrared (LWIR) absorption band of methane. The most used bands for methane detection are the near-infrared (NIR) and medium-wavelength infrared (MWIR). It is related to the price and availability of components as well as the location of the characteristic absorption bands of many gases with a high absorption coefficient (so-called molecular fingerprint regions). However, the use of other bands may also have some advantages in gas detection. This paper concerns the study of the methane sensor, based on an 8.014 µm absorption line, due to its high and untapped application potential [21]. The selected spectral line has a direct impact on the laser operating point, but its width, i.e., full width at half maximum (FWHM), should also be considered as it depends on the pressure, temperature, and methane concentration [12].

In the experimental setup, the distributed feedback (DFB) type QCL 8.0 μm laser (AdTech Photonics, City of Industry, CA, USA) was chosen as the infrared source for the methane sensing. The schematic diagram of the experimental setup used in this work is depicted in Figure 1a. Within the chosen laser’s tuning range, absorption lines of CH_4_ particles exist. At a pressure of 100 mbar, two single absorption lines can be observed (at 8013.43 nm and 8014.71 nm), as well as a stronger double line at 8013.96 nm (Figure 1b). However, the double line produces distorted harmonics signals, which can make the measurement more difficult in WMS setup [24]. Therefore, it was decided to choose a single absorption line at 8014.71 nm.

The temperature of the laser was stabilized to 39.07 °C using a TEC 3210 (Arroyo Instruments, San Luis Obispo, CA, USA). Current control was performed by an LDX-3232 current source (ILX Lightwave, Newport Corporation, Irvine, CA, USA), and the laser operating point was set to 275 mA. Tuning and modulation of the wavelength was carried out by applying voltage signals (ramp and sinusoid) to the input of the current controller from the AFG 3252 generator (Tektronix, Beaverton, OR, USA). Since the LDX-3232 has only one input for the modulation signal, a low-noise signal adder was made. In addition to the function of summing the voltage of the ramp and sine signals, the adder was also equipped with band-pass filters, whose frequency response is limited to 5–5000 Hz for the ramp signal input and 5–60 kHz for the sine signal input, respectively. Setting the voltage values of the ramp and sine signals takes place in the AFG 3252 generator, while the adder uses a 20 dB attenuator of the sine signal voltage, pre-setting the proportions between the signals. This allowed the generation of a sine signal in the AFG 3252 generator with a larger amplitude. All these procedures contributed to the improvement of the ratio of the signal voltage to the noise voltage fed to the input of the laser controller.

As an absorption cell, an HC30L/M-M02 Herriott-type multipass cell (Thorlabs, Newton, MA, USA) providing 31.2 m optical path (81 internal reflections of the laser beam) was applied. To collimate the laser beam from the QCL, an aspheric lens (Lens 1) C093TME-F, f = 3.0 mm, NA = 0.71 (ThorLabs) was used. A spatial filter was used to appropriately focus the laser beam in the middle of the multipass cell and to minimize interferences. The filter consisted of two CaF_2_ Plano-Convex Lens 2, 3 with focal lengths 50 mm and 100 mm and ring-actuated iris diaphragms SM1D12CZ (ThorLabs). There were also two folding mirrors prior to the Herriott cell to position and angle the incident beam. Optical signals were detected at the output of the multipass cell using a CaF_2_ plano-convex lens f = 20 mm and PCI-4TE-9-2x2 photodetector (VIGO Photonics, Ożarów Mazowiecki, Poland). Afterward, signals from the photodetector were digitized using a PXIe-5172 oscilloscope card, which was controlled using a PXIe-8880 controller (National Instrument, Austin, TX, USA). Software for data acquisition, processing, and system control was implemented using a script in MATLAB R2022b (MathWorks, Natick, MA, USA). A custom lock-in software amplifier was used to demonstrate harmonics. The developed system also enables automatic control of the AFG 3252 generator using the VISA interface and the GPIB link between the generator and the NI PXI-8232 card (National Instrument). In the experiment, the wavelength of the QCL was periodically tuned by a ramp signal from the generator to sweep the selected methane absorption line. Additionally, the laser beam was modulated with a sinusoidal signal. Considering frequency bandwidths and assumed parameters ranges of devices based on literature analysis, the following modulation parameters were initially selected: modulation frequency—40 kHz, sine-wave amplitude 150 mV_pp_, ramp frequency—200 Hz, ramp amplitude 300 mV_pp_, sampling frequency 10 MHz. The initial settings were implemented based on the characteristics of the system components and assumptions from literature references. They represent the sensor’s initial operating conditions parameters prior to the implementation of optimization techniques. This approach reflects the procedure experts followed during the development of gas sensors. Prior to the final optimization of parameters, the initial selection of parameters is coarse and characterized by a certain degree of randomness. Subsequently, parameter fine-tuning is conducted to achieve the best results. Considering the accuracy of the devices used, their adjustment resolution, and assumed maximum ratings, approx. 1.39 × 10^32^ of settings changes and measurements are required to perform the WMS sensor characterization in selected operating ranges and with the maximum accuracy achievable with the devices used. This number was determined based on the possible combinations of device settings listed in Table 1. Assuming this is a tremendous number, the total time required to conduct such complex experiments may be hundreds of hours. Certainly, conducting sensor parameter measurements solely at specific measurement points is feasible; however, this necessitates providing expert knowledge and ensuring the inclusion of crucial regions to prevent any appropriate solution omissions. Therefore, in the pursuit of automating the process, an optimization algorithm may be employed to address the issue of investigating many combinations.

### 2.2. Test the Sensor Operation with Initial Settings

The multipass cell was filled with the ambient air and a 100% CH_4_ mixture under the pressure of 100 mbar reached by a vacuum pump. The output signal from the photodetector was converted to digital form with a 10 MHz sampling rate and was demodulated using a software phase-sensitive amplifier. Figure 2 shows an example of the first four harmonics recorded by the system, as well as their SNR values. SNR was defined as the ratio of the maximal amplitude of the harmonic signal to the noise RMS from the non-absorbing part of the scan. As a result of the tests, the following SNR of harmonics were obtained without averaging: 1f—214, 2f—46, 3f—13, 4f—6. They are reference values for the evaluation of the proposed method.

### 2.3. Optimization Procedure

Genetic algorithms (GA) were proposed to optimize the operation of the described sensor. GA is part of the evolutionary algorithms that belong to the branch of artificial intelligence. This type of metaheuristic was developed by J.H. Holland, who was inspired by observations of biological evolutionary processes to solve optimization problems [25]. Genetic algorithms are metaheuristics that enable a time-efficient search for near-optimal solutions based on biologically inspired mechanisms such as selection, crossover, and mutation. This type of algorithm offers several advantages over traditional optimization methods. Firstly, unlike classical optimization gradient methods, population-based algorithms do not require computationally complex derivative calculations. Genetic algorithms are effective in solving optimization problems with a large number of variables and for noisy, complex, or even discrete objective functions. Additionally, crossover and mutation operations efficiently reduce the risk of getting stuck at a local minimum. Moreover, employing genetic algorithms enables the development of hybrid optimization tools by integrating multiple algorithms, thereby enhancing optimization efficiency. GA has been successfully used in many fields of science, including laser absorption spectroscopy, where genetic algorithms were used to optimize the construction of multi-pass cell mirrors [26] or for improving the concentration inversion precision of the system extreme learning machine [27]. The advantages of GAs and an increasing number of solved various problems by metaheuristics encourage the use of these algorithms in WMS research.

During the experiment, modulation parameters of the constructed WMS setup were optimized using the function ‘ga’ provided using MATLAB R2022b software. This function can solve both constrained and unconstrained optimization problems based on a natural selection process. The operation of the algorithm begins with the initialization phase, which consists of randomly generating the first population in linear constraints. In this case, each individual contained information regarding WMS sensor modulation parameters. The population was sequentially implemented in the system by sending VISA-GPIB commands to change the generator operation points. This process included time for the system to stabilize. The usefulness of the properties of each candidate (chromosomes) was assessed using the fitness function that was defined as a negated value of the signal power to noise power ratio of WMS harmonic. This operation was conducted using software Lock-in, harmonic peak-detector, and standard deviation calculations. The phase-sensitive detection was realized by a multiplication signal obtained at the photodetector output within one laser scan by sine and cosine waves with a modulation frequency or its higher harmonics. Subsequently, the received in-phase and quadrature signals were filtered by low-pass digital filters. The resulting Lock-In output signal was derived from the magnitude of these two components. In the next step, the part of the population was selected using an appropriate selection function (e.g., roulette wheel or tournament) for further processing. The algorithm repeatedly modified a population of individual solutions. At each step, the genetic algorithm used selected individuals from the current population as parents to produce the children for the next generation. Moreover, individuals were additionally mutated to maintain the genetic diversity of the chromosomes. After selection, recombination, and mutation operation, a new generation was evaluated. The task of the algorithm was to find the global minimum of the fitting function defined as the negated SNR value, also known as the objective function. This algorithm was repeated iteratively until the set of final conditions was met. Over successive generations, the population evolved toward an optimal solution. Additionally, solutions that exceeded the constraints were not generated in the system to prevent damage to the laser. The flowchart of the optimization algorithm is depicted in Figure 3.

The proposed solutions were aimed primarily at automating the process of searching for optimal modulation parameters without expert knowledge, the WMS mathematical model, or human assistance. Moreover, the utilization of metaheuristics addresses the challenge of avoiding local minima and decreases the necessity to explore numerous solutions, consequently reducing the time required for optimization. The ‘ga’ parameters and lower and upper constraints of the search for solutions by the genetic algorithm are summarized in Table 2. The population size and number of generations were experimentally chosen to provide a solution convergence in as few operations as possible.

The crossover operation in the proposed algorithm was executed using the ‘crossoverintermediate’ function, which generates children by computing a weighted average of the parents’ values. Throughout the experiments, the crossover operation was implemented with equal weighting for all parents. Additionally, the mutation operation utilized the ‘mutation power’ function from MATLAB, employing default hyperparameters. This function computes mutated solutions by adding or subtracting a random value multiplied by the difference between the component value and its lower constraint. Moreover, four different selection methods, provided using MATLAB, were tested during the research: selectionstochunif, selectionremainder, selectionroulette, and selectiontournament. Applied constraints considering electrical parameters of system devices allowed the search area to be narrowed down and components protected from damage. In the above-described way, evolutionary optimization has been achieved with the developed script that controlled real-time whole WMS system.

## 3. Results and Discussion

The optimization procedure for 20 generations lasted about 300 s, including the implemented delay aimed at stabilization of the laser operating point before the digital signal processing. The termination condition was determined experimentally as the number of 20 generations. Running the algorithm for a larger number of generations did not lead to a significant improvement in SNR and only increased the computation time. The hardware had the following specifications: CPU Intel Xeon 2.30 GHz, 24 GB RAM. The algorithm’s running time is for sure much shorter than the time it would take for a specialist to find a suitable solution to this problem or check all possible system settings by brute-force search. In each iteration of the algorithm, 400 solutions (20 individuals and 20 generations) were checked, and the best individual parameters were recorded. This number is relatively low compared to the number of possible settings of key system elements and capabilities of non-heuristic optimization methods in five dimensions solution space. Figure 4 and Figure 5 show plots of mean scores of the GA optimization process for the first four harmonics.

Optimization of modulation parameters using genetic algorithms improved the SNR for each harmonic relative to the initial settings. Table 3 shows the best optimization results for the first four harmonics, based on the individual with the highest fitness value from all generations. The maximum improvement was achieved at 360% for the first harmonic, 167% for the second harmonic, 346% for the third harmonic, and 650% for the fourth. This emphasizes the importance of the optimization process, especially for higher harmonics. Given the stochastic nature of the algorithm, each run may yield slightly different results, which is consistent with the fact that metaheuristics determine near-to-the-optimal estimation but not always the best possible. This research did not show a clear advantage of any selection function for WMS modulation parameters optimization; however, the use of several functions allowed us to assess the convergence of the algorithm’s solution. Even and odd harmonics of higher order (3f and 4f) showed better optimization convergence than 1f or 2f. For all harmonics, the modulation amplitude had the smallest statistical dispersion out of the encoded best individual’s modulation parameters.

This confirms the existence of the global maximum of SNR related to the modulation index and the importance of the correct selection of the modulation signal amplitude. Although the statistical dispersion of the modulation frequency for the best individuals was large, for all cases, it was greater than 20 kHz. This fact is probably related to the minimization of 1/f noise for higher modulation frequencies. On the other hand, the algorithm did not choose the maximum value related to the adopted search constraints, probably due to the frequency response bandwidths of elements in the system. The statistics of the optimization process are shown in Figure 6.

The frequency of the ramp signal was also characterized by a large statistical dispersion. However, it was in the range of 75 Hz (lower whisker) to 777 Hz (upper whisker), excluding values at the boundaries of the search area. These frequencies provide a high ratio of modulation frequency to ramp frequency, usually greater than 100. Additionally, the median value prompts the selection of ramp tuning to lower frequencies. Finally, ramp amplitude optimization was analyzed. Assuming that there are no other interfering lines in the vicinity of the detected gas line, optimizing the selection of this parameter is easier to perform. However, the evaluated median value shows that the range of tuning should be limited to the vicinity of the absorption line. The results of the conducted modulation parameters optimization are consistent with the information presented in the literature [9,11,17]. Due to the complex intercorrelation between modulation parameters, there is no explicit universal recipe for selecting the WMS modulation parameters. The right selection depends on the individual sensor elements and gases being detected. However, the use of metaheuristic genetic algorithms allows for the precise selection of modulation parameters among a huge number of combinations of all sensor component settings in order to improve its SNR in a very short time. As a result of the proposed algorithm, a 3.6-fold improvement in the SNR of the 1f signal was obtained, and from (4), it can be calculated that the LOD can be reduced by a factor of 0.28. Similarly, for harmonics 2f, 3f, and 4f, *C_L_* will be decreased by factors of 0.58, 0.29, and 0.15, respectively.

## 4. Conclusions

In this paper, application of genetic algorithms to solve the issue of selecting modulation parameters for wavelength modulation spectroscopy was reported for the first time. The proposed approach optimizes the modulation parameters of the WMS gas sensor by combining remote control of individual sensor components, digital signal processing, and an AI algorithm. This solution will allow automating the process of parameter tuning and data analysis, reducing the need for manual calibration and speeding up the experimentation process. It can also be directly used to make fine adjustments during the measurement process in response to changing measurement conditions, addressing critical domains such as safety, industry, and environmental monitoring. In this case, the developed algorithm can adapt the modulation parameters in real-time to ensure consistent and reliable measurements. It should be emphasized that the optimization of WMS sensor parameters was conducted in real experiments using the developed methane sensor rather than solely using computer simulation, which could lead to overlooking certain technical aspects of the equipment. This confirms the usefulness, robustness, and reliability of the proposed solution. Future research will involve exploring other metaheuristic algorithms and selecting their hyperparameters to enhance the performance of WMS sensors. The impact of employing alternative approaches on optimization time, convergence, and reliability in practical applications will also be investigated.

## Figures and Tables

**Figure 1 sensors-24-01842-f001:**
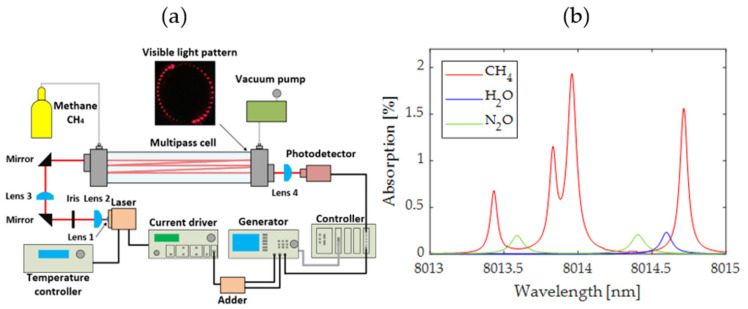
(**a**) Schematic of the CH_4_ detection system; (**b**) HITRAN simulation results (1.8 ppm CH_4_ absorption spectrum at temperature 296 K, pressure of 100 mbar, pathlength 31.23 m with air absorption bands of 7% H_2_O and 322 ppb N_2_O).

**Figure 2 sensors-24-01842-f002:**
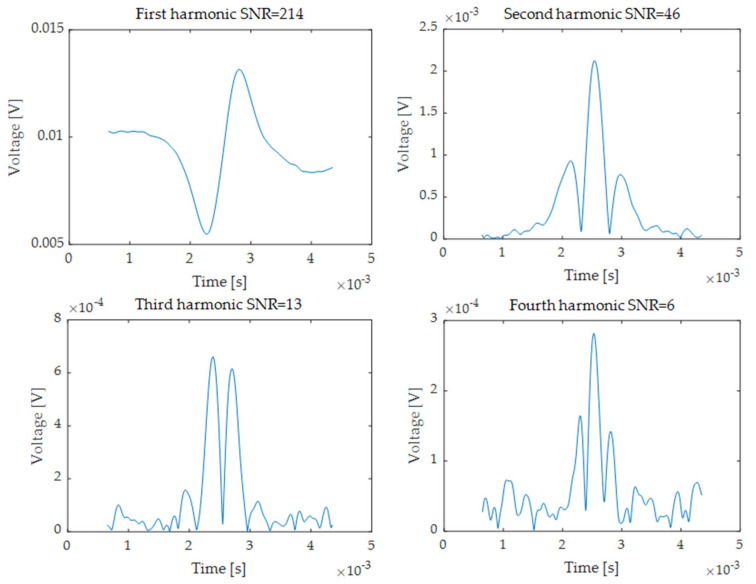
Measured first four WMS harmonics signals with SNR value.

**Figure 3 sensors-24-01842-f003:**
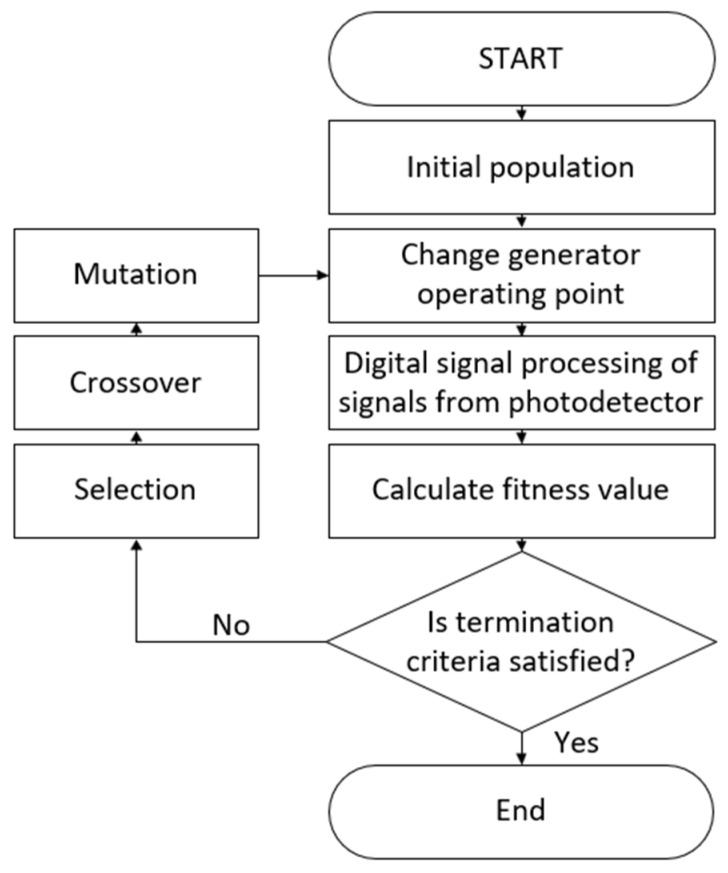
The flowchart of the optimization algorithm.

**Figure 4 sensors-24-01842-f004:**
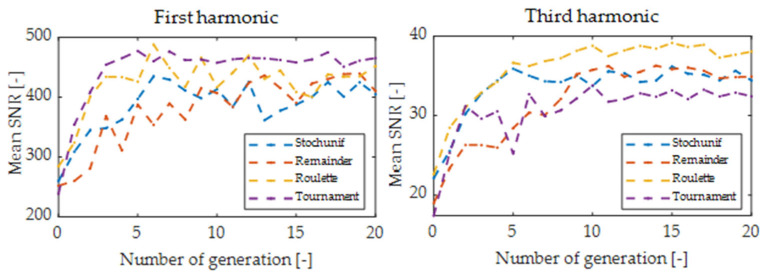
Mean scores versus generation for odd harmonics with various selection functions.

**Figure 5 sensors-24-01842-f005:**
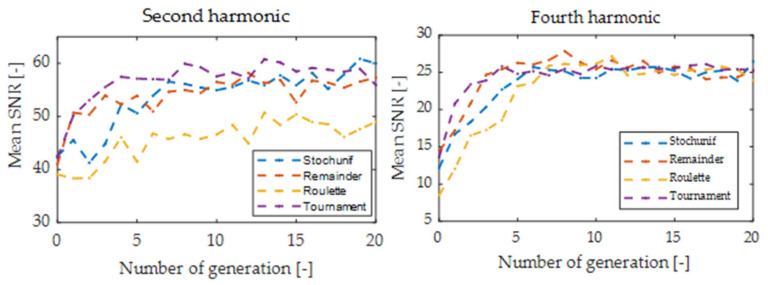
Mean scores versus generation for even harmonics with various selection functions.

**Figure 6 sensors-24-01842-f006:**
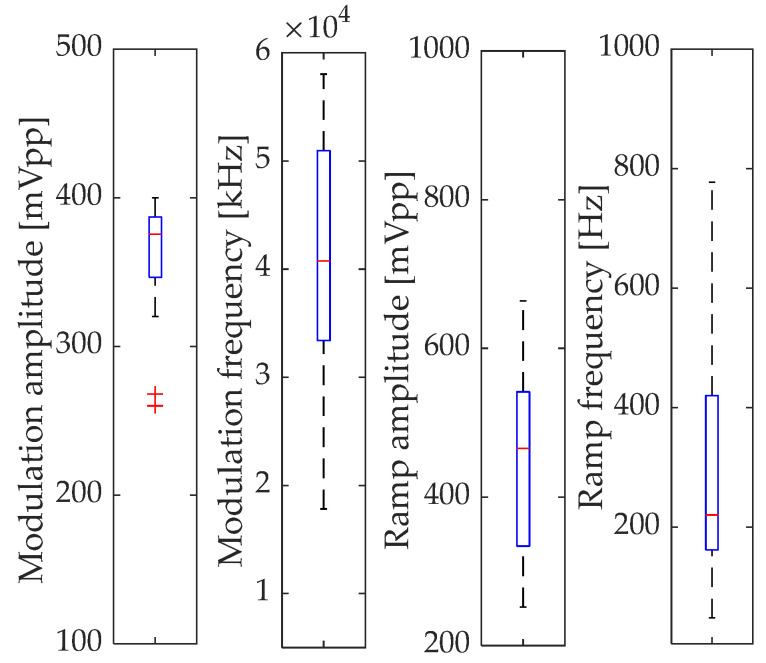
Box plots of best individuals for GA optimization. Central mark indicates the median, and the bottom and top edges of the box indicate first and third quartile. The whiskers extend to the most extreme data points, and the outliers are plotted individually using the ‘+’ marker symbol.

**Table 1 sensors-24-01842-t001:** Number of possible settings of key system elements.

Device	Parameter	Accuracy	Resolution	Range of Research	Number of Points	Remarks
LDX 3232	Laser current	±0.15% of setpoint ± 2 mA	40 µA	245–525 mA	7k	Laser wavelength tuning: 0.065 nm/mA
TEC 3210	Laser temperature	0.05 °C	0.01 °C	14–45 °C	3k	Laser wavelength tuning: 0.638 nm/°C
AFG 3252	Modulation frequency	±1 ppm ± 1 μHz	1 μHz	5–60 kHz	55M	-
	Modulation amplitude	±(1% of setting + 1 mV)	0.1 mV_pp_	100–500 mV_pp_	4k	Laser current driver conversion: 200 mA/V
	Scanning frequency	±1 ppm ± 1 μHz	1 μHz	5–5000 Hz	4M	-
	Scanning Amplitude	±(1% of setting + 1 mV)	0.1 mV_pp_	250–1000 mV_pp_	7.5M	Laser current driver conversion: 200 mA/V

**Table 2 sensors-24-01842-t002:** Parameters of the optimization algorithm.

Parameter	Value
Population size	20
Number of generations	20
Creation function	Gacreationlinearfeasible
Crossover function	Crossoverintermediate
Mutation function	Mutationpower
Sine-wave amplitude constraints	100 mV_pp_–500 mV_pp_
Modulation frequency constraints	5 kHz–60 kHz
Ramp amplitude constraints	250 mV_pp_–1000 mV_pp_
Ramp frequency constraints	5 Hz–5000 Hz

**Table 3 sensors-24-01842-t003:** The best individuals of GA even and odd harmonics optimization.

Selection Function	Modulation Amplitude [mV_pp_]	Modulation Frequency [kHz]	Ramp Amplitude [mV_pp_]	Ramp Frequency [Hz]	SNR [*v*/*v*]	SNR Improvement [%]
The first harmonic						
Stochunif	376	47,272	530	467	770	360
Remainder	393	58,031	539	777	750	350
Roulette	344	48,323	339	252	700	327
Tournament	393	33,930	492	161	580	271
The second harmonic						
Stochunif	349	53,599	664	48	77	167
Remainder	260	36,821	544	96	73	159
Roulette	268	44,631	585	513	74	161
Tournament	359	58,036	445	216	75	163
The third harmonic						
Stochunif	385	36,927	329	276	45	346
Remainder	376	55,205	326	163	44	338
Roulette	388	46,617	576	75	45	346
Tournament	320	33,965	265	188	44	338
The fourth harmonic						
Stochunif	367	32,844	446	206	34	567
Remainder	386	32,296	483	398	37	617
Roulette	400	24,768	252	442	39	650
Tournament	375	27,819	448	225	34	567

## Data Availability

The data presented in this study are available on request from the corresponding author.

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
