# Peer review of "Time-Efficient SNR Optimization of WMS-Based Gas Sensor Using a Genetic Algorithm"

_sensors, 2024, doi:10.3390/s24061842_

Round 1
Reviewer 1 Report
Comments and Suggestions for Authors
F. Musiałek et al. introduce in this work a classical optimization approach using the genetic algorithm for optimizing parameters in wavelength modulation spectroscopy experiments. The study demonstrates a substantial 1.6 to 6.5 times improvement in signal power to noise power ratio depending on the harmonic. The proposed methodology, equipped with a custom-made experimental setup, notably reduces the time required to identify optimal settings, achieving efficiency in exploring a wide range of WMS parameter combinations.
1) How does the paper contribute to the existing knowledge in the field of gas sensor technology, particularly in terms of the innovative use of genetic algorithms for optimizing the signal power to noise power ratio (SNR) in WMS-based gas sensors?
2) Are the experimental results and methodology presented in the paper robust and reliable, and do they convincingly support the claim that the proposed genetic algorithm approach can significantly reduce the time needed to identify optimal settings for WMS-based gas sensors?
3) Why the authors have not used classical optimization approach while considering bio-inspired optimization method. Is it realy needed.
4) Is the obtained combination reflects the best of the best scenario
5) How does the paper address potential limitations or challenges associated with the application of genetic algorithms and AI algorithms in the context of gas sensor optimization? Are there alternative approaches or considerations that the paper could discuss such as ant colony optimization, PSO, ..etc? Why you have choosen the GA-based optimization approach.
6) The objective function versus the generation should be plotted while showing the convergeance apect.
7) In terms of practical applications, does the paper discuss how the proposed approach could impact real-world scenarios, such as industrial gas sensing applications or environmental monitoring? Additionally, are there any potential scalability issues or considerations for implementing the proposed methodology in different contexts?
Considering the above comments, the manuscript can be accepted after minor revision.
Good luck.
Comments on the Quality of English LanguageN/A
Reviewer 2 Report
Comments and Suggestions for Authors
Your manuscript proposes the application of genetic algorithms to solve the issue of selection modulation parameter for wavelength modulation spectroscopy, It has realized the optimization of modulation parameters of WMS gas sensor, which is innovative to some extent, but there are still some problems in the manuscript, so the following suggestions are put forward to you:
1、In the genetic algorithm, in order to maintain diversity in the search process and not fall into the local optimal solution prematurely, it is necessary to reasonably design crossover and mutation operations. The specific processes of these two operations are not introduced in your manuscript, so it is suggested to add.
2、In the manuscript, what are the specific termination conditions to judge whether the genetic algorithm has converged to a better solution?
3、The manuscript mentions the use of software lock-in, harmonic peak-detector and standard deviation calculations to realize the use of fitness function to evaluate the usefulness of each candidate attribute, what is the specific operating process? How is this achieved?
4、In the manuscript, you compare the experimental results of the proposed genetic algorithm with those of the initial setting, thus verifying the advantages of the proposed genetic algorithm. However, the basis for the initial parameter setting in the experiment is not explained in detail, so it is suggested to add.
